# Reliable Noninvasive Glucose Sensing via CNN-Based Spectroscopy

El Arbi Belfarsi*, Henry Flores†, and Maria Valero†

*Department of Computer Science, †Department of Information Technology

Kennesaw State University, Georgia, USA

0009-0006-4145-5492, 0009-0005-6360-9977, 0000-0001-8913-9604

*Abstract*—In this study, we investigate two distinct methods based on short wave infrared (SWIR) spectroscopy for non-invasive glucose monitoring. The first method employs a multi-wavelength SWIR imaging system combined with convolutional neural networks (CNNs) to extract spatial features associated with glucose absorption. The second method utilizes a compact photodiode voltage sensor with machine learning regressors (e.g., random forest) applied to normalized optical signals. Both methods were evaluated on synthetic blood phantoms and skin-mimicking materials across physiological glucose concentrations (70–200 mg/dL). The CNN-based system achieved a mean absolute percentage error (MAPE) of 4.82% at 650 nm, with 100% Zone A coverage in the Clarke Error Grid. In contrast, the photodiode system achieved 86.4% Zone A accuracy. Together, these results demonstrate distinct sensing strategies that balance clinical accuracy, cost considerations, and potential for wearable integration, contributing to the development of reliable continuous non-invasive glucose monitoring.

*Index Terms*—Noninvasive glucose monitoring, spectroscopy, convolutional neural networks, random forest regression, photo-diodes, feature extraction, wearable sensors.

## I. INTRODUCTION

Diabetes mellitus affects over 536 million adults worldwide (10.5% of adults) as of 2021, and is projected to reach 783 million by 2045 [1]. Maintaining blood glucose within target ranges is essential to prevent complications such as neuropathy, retinopathy, and cardiovascular disease [2]. Frequent self-monitoring of blood glucose (SMBG) via finger-prick tests remains the standard approach; however, it is invasive, painful, and can reduce patient adherence by over 63% of the recommended levels [3]. Continuous glucose monitoring (CGM) systems mitigate some discomfort by providing near real-time readings but still require subcutaneous sensors, carry high costs, and can suffer accuracy drift during rapid glycemic changes [4], [5].

The invasiveness and expense of current methods limit frequent monitoring, especially in resource-constrained settings. A truly noninvasive glucometer would enhance patient compliance and quality of life by offering painless, affordable, and convenient measurement [5]. Despite decades of research, no technology has yet achieved the accuracy required to replace SMBG or CGM [5].

This work was supported in part by the National Institute On Aging of the NIH under Award Number P30AG073105. The content is solely the responsibility of the authors and does not necessarily represent the official views of the NIH. This work was also partially supported by the Georgia Research Alliance under Award Number GRA.25.016.KSU.01.a.

Optical methods dominate noninvasive research [4]. Near-infrared (NIR) spectroscopy (750–2500 nm) can penetrate skin and interact with glucose absorption bands but suffers from weak glucose signals overlapped by water and protein peaks [6], [7]. Table I summarizes these overlapping absorption features [8].

TABLE I: NIR absorption peaks of key biological components [8].

| S.no | Component | Absorption Peaks (nm) |
|---|---|---|
| 1 | Glucose | 1408, 1536, 1688, 2261 |
| 2 | Water | 1450, 1787, 1934 |
| 3 | Fat | 2299, 2342 |
| 4 | Protein | 2174, 2288 |

Multivariate calibration or machine learning is required to isolate glucose signals [7], [9]. Factors such as skin thickness, hydration, and temperature add variability [5]. Mid-infrared (MIR) offers stronger glucose absorbance but poor penetration [6]. Raman spectroscopy provides better specificity but weak signals and system complexity limit clinical feasibility [4]. Other techniques (electromagnetic sensing, polarimetry, photoacoustic spectroscopy) face challenges in signal-to-noise ratio, tissue heterogeneity, and calibration [10].

The shortwave infrared (SWIR) region (700–2000 nm) shows promise: it penetrates deeper with reduced scattering and includes glucose overtone bands with less water interference, notably around 1600 nm [6], [17]. Hong *et al.* demonstrated that 1500–1700 nm yields improved imaging depth and contrast in biological tissues [17].

We benchmark two ML-driven techniques for noninvasive glucose sensing, validated on *in vitro* samples within the physiological range of 70–200 mg/dL:

- **SWIR Imaging Technique:** Multi-wavelength SWIR imaging using 650 nm, 808 nm, and 850 nm lasers, along with an 850 nm LED with 50 mW power. Deep neural networks predict glucose concentration directly from the captured images.
- **Voltage Technique:** SWIR sensing at 1600 nm using a 2 mW LED source and an InGaAs photodiode. Voltage ratios (post-beam to pre-beam) are analyzed using linear regression (LR), multiple linear regression (MLR), and random forest (RF) models.

Both techniques utilize synthetic skin and blood phantoms designed to replicate physiological optical properties. Subse-

TABLE II: Summary of Related Work: Approach, Model, RMSE, and CEG Zone A Coverage.

| Reference | Approach | Model | RMSE (mg/dL) | Zone A (%) |
|---|---|---|---|---|
| Cistola et al. [11] | SWIR Imaging (1500–1700 nm) | Random Forest Regression (RFR) | 10.9 | 93.1 |
| Javid et al. [12] | NIR Spectroscopy (700–1100 nm) | LR / MLR | 14.8 | 80.0 |
| Aloraynan et al. [13] | Mid-IR Photoacoustic | Ensemble ML Classifiers | 25 mg/dL | N/A |
| Abdolrazzaghi et al. [14] | Split-Ring Resonator (1.156 GHz) | Resonator Sensing | N/A | N/A |
| Zeynali et al. [15] | PPG Signals | ResNet34-1D | 19.7 | 76.6 |
| Anis and Alias [16] | Portable NIR (900–1600 nm, IoT) | IoT Regression (Correlation) | N/A | N/A |

quent sections detail the experimental setup, data acquisition process, and AI-based analysis methods.

## II. RELATED WORK

Noninvasive glucose monitoring approaches can be broadly categorized into optical imaging/spectroscopy, electromagnetic/resonator sensors, and wearable/IoT devices. Below, we review six representative studies—strictly limited to those provided—and summarize their reported performance metrics.

### A. Optical Imaging and Spectroscopy

Cistola et al. [11] employed SWIR imaging (1500–1700 nm) on synthetic skin and blood phantoms across concentrations of 50–200 mg/dL. Using Random Forest Regression on pixel-wise reflectance features, they achieved an RMSE of 10.9 mg/dL with 93.1% of predictions in Zone A and 6.9% in Zone B of the Clarke Error Grid.

Javid et al. [12] proposed a smartphone-based NIR spectroscopy system (700–1100 nm) operating in reflectance and transmittance modes. Linear and multiple linear regression were applied to five discrete glucose levels (50–250 mg/dL), yielding an RMSE of 14.8 mg/dL and 80% Zone A coverage. However, when using finer 10 mg/dL increments, RMSE rose above 18 mg/dL and Zone A coverage dropped below 70%, indicating reduced linear generalization.

Aloraynan et al. [13] used mid-IR photoacoustic spectroscopy with a quantum cascade laser and artificial skin phantoms. Ensemble machine learning models achieved a sensitivity of 25 mg/dL across the physiological range, though no CEG analysis was reported.

### B. Electromagnetic and Resonator Sensors

Abdolrazzaghi et al. [14] developed a 1.156 GHz split-ring resonator with loss compensation, increasing the quality factor from 190 to 3850. In aqueous and serum-like glucose solutions (1–30 mM), they demonstrated a linear frequency shift with a detection limit of 18 mg/dL. However, neither RMSE nor Clarke Error Grid metrics were provided.

Zeynali et al. [15] applied deep learning to PPG data from the VitalDB dataset (699,000 segments from 6,368 patients). Among ResNet34-1D, VGG16-1D, and CNN-LSTM models, the best-performing model, ResNet34-1D, achieved an RMSE of 19.7 mg/dL with 76.6% in Zone A and a 6.4 s inference time on TinyML hardware.

Anis and Alias [16] introduced a handheld noninvasive glucometer using NIR spectroscopy (900–1600 nm) with IoT-based data transmission. They reported a correlation coef-

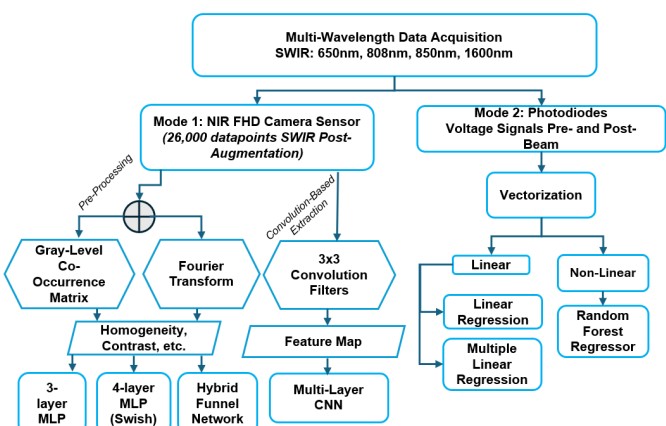

Fig. 1: Block diagram representing glucose prediction approaches

ficient of 0.85 against an invasive reference device, though RMSE and CEG metrics were not disclosed.

**Summary and Benchmark Selection** Table II summarizes RMSE and Zone A results from prior studies. Among them, Cistola et al. [11] provides the most realistic experimental setup and best overall performance. Their work serves as the primary benchmark for evaluating our system.

## III. METHODOLOGY

### A. Overview

Fig. 1 showcases the overall methodology followed in the paper. In this study, we implemented two complementary modes of data acquisition to investigate non-invasive glucose estimation: (i) SWIR imaging, and (ii) photodiode-based voltage acquisition. This section details the methodology employed for each approach.

### B. Mode 1: SWIR Imaging Dataset

*1) Experimental Setup:* Fig. 2 shows a custom 3D-printed black cuvette holder maintained consistent alignment. A 3.6 mm fully high-resolution (FHD) SWIR camera was mounted on one side, while four light sources (650 nm laser, 808 nm laser, 850 nm laser, and 850 nm LED) provided transillumination from the opposite side. Two silicone-based skin phantoms, each 3 mm thick and containing embedded scattering particles, were positioned at the illumination and detection sides of the sample. This setup was designed to closely replicate the optical scattering characteristics of human subcutaneous tissue, enhancing the physiological relevance of the experiment.

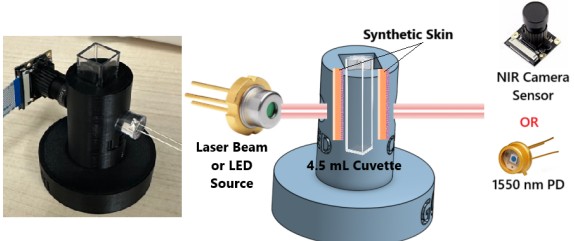

Fig. 2: Optical measurement setup using synthetic skin model with interchangeable NIR camera or photodiode sensor.

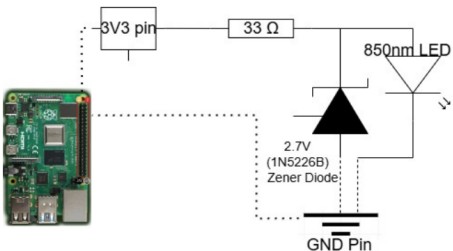

Fig. 3: Circuit design for driving the LED/Laser using a 2.7 V Zener diode and 33 Ω resistor regulated by a 3.3 V supply.

The LED circuit (Fig. 3) consists of a 1.5 V, 50 mA LED in series with a 33 Ω resistor, powered by the Raspberry Pi's 3V3 and GND pins. A 2.7 V, 500 mW Zener diode (1N5226B) is connected in parallel with the LED to regulate voltage and suppress transients [18].

*2) Sample Preparation:* Glucose solutions were prepared in-vitro by diluting concentrated glucose stock solutions (700–2000 mg/dL) to physiological ranges of 70–200 mg/dL using a 1:10 dilution in synthetic blood concentrate. The glucose used was in the form of dextrose, which accounts for approximately 99% of the glucose found in the human body. The synthetic blood concentrate consisted of a mixture designed to mimic human blood optical properties, including water, salts (e.g., sodium chloride), proteins (such as albumin), and hemoglobin analogs or red dye compounds to replicate absorption and scattering characteristics. Concentration steps were incremented by 2 mg/dL, ensuring fine-grained dataset diversity.

*3) Data Acquisition and Augmentation:* For each wavelength and glucose concentration, 10 SWIR images were captured, resulting in a raw dataset of NIR images. To enhance model generalization and simulate real-world variances, data augmentation was applied, including:

- Randomized contrast alterations,
- Image rotations,
- Application of Gaussian noise.

The final dataset consisted of approximately 6500 datapoints after augmentation, used for downstream model development and evaluation.

*4) Feature Extraction: Texture and Frequency Domain Analysis:* To enhance the model's ability to capture complex patterns present in the SWIR images, we applied both spatial and frequency domain feature extraction techniques. These transformations are designed to extract robust descriptors that may correlate with glucose-induced optical variations [19].

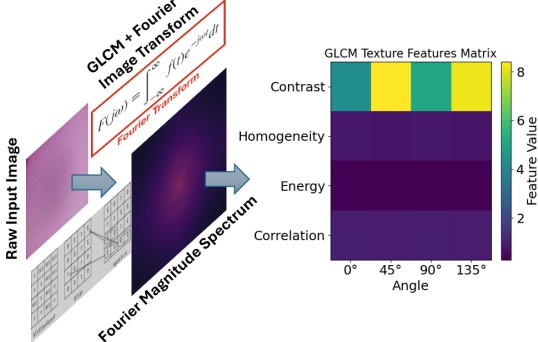

Fig. 4: Feature extraction using Fourier transform and GLCM texture analysis on the input image.

*a) Gray Level Co-Occurrence Matrix (GLCM):* The Gray Level Co-Occurrence Matrix (GLCM) quantifies texture by measuring how often pairs of pixel intensities occur at a specified offset and orientation. For a grayscale image $I(x, y)$ of width $W$ and height $H$, quantized to $G$ gray levels, the GLCM $P(i, j, d, \theta)$ counts the occurrences where a reference pixel with intensity $i$ has a neighboring pixel with intensity $j$ at distance $d$ and angle $\theta$.

These features capture the micro-structural organization of the optical patterns, which may vary with glucose concentration due to changes in scattering and absorption properties [19].

*b) Fourier Transform Features:* In parallel, the 2D Discrete Fourier Transform (DFT) captures the frequency-domain representation of the image $I(x, y)$ of size $W \times H$. The transformed spectrum $F(u, v)$ is computed as:

$$F(u, v) = \sum_{x=0}^{W-1} \sum_{y=0}^{H-1} I(x, y) e^{-j2\pi \left( \frac{ux}{W} + \frac{vy}{H} \right)}$$

where $(u, v)$ are the frequency coordinates corresponding to the spatial coordinates $(x, y)$.

From the magnitude spectrum $|F(u, v)|$, we extracted:

- Low-frequency energy (central concentration of energy),
- High-frequency content (edge and fine texture variations),
- Spectral entropy.

Frequency domain analysis assists in characterizing periodic patterns and subtle variations not easily observable in spatial domain alone [20].

*c) Feature Fusion:* The extracted GLCM and Fourier features were concatenated to form a composite feature vector, which was subsequently fed into the regression model alongside the raw image inputs. This hybrid approach was chosen to combine handcrafted features that capture domain-specific patterns with the learning capacity of the CNN. Texture features derived from GLCM can highlight subtle spatial

variations in image intensity caused by glucose-induced optical changes, while Fourier features capture global frequency patterns related to light scattering and transmission properties. Such features have been shown to enhance performance in biomedical imaging tasks, including glucose-related analysis [21]. By integrating these features with raw image data, the model leverages both explicitly engineered descriptors and automatically learned representations for improved glucose estimation accuracy. In Fig. 4, the feature extraction pipeline illustrates the sequential computation of the Fourier magnitude spectrum and GLCM texture metrics, which jointly contribute to the model's predictive performance.

### C. Mode 2: Photodiode Voltage Readings

*1) Experimental Setup:* To adequately compare the SWIR imaging approach, a simplified hardware configuration was developed by replacing the IR camera with a photodiode receiver. The updated setup consists of:

- **LED Source:** A 1600 nm LED was selected as the illumination source, targeting absorption peaks of glucose molecules in the (SWIR) spectrum.
- **Photodetector:** A 1550 nm photodiode receiver positioned opposite the LED to collect transmitted light intensity.

The rationale for selecting this configuration includes: (i) enabling more compact prototype designs suitable for mobile health applications, (ii) conducting reproducibility studies to validate findings from the imaging dataset, and (iii) exploring the underrepresented 1600 nm wavelength range, which remains relatively less investigated in the literature for noninvasive glucose sensing.

*2) Voltage Dataset Acquisition:* For each sample, baseline voltage readings were first recorded under no-beam conditions to account for system offsets. Subsequently, voltage measurements were collected both before and after the beam interaction with the glucose sample:

- $V_{\text{baseline}}$: baseline voltage without illumination.
- $V_{\text{pre}}$: voltage recorded prior to beam activation.
- $V_{\text{post}}$: voltage recorded after beam transmission through the glucose solution.

The differential voltage readings captured light attenuation and scattering effects as a function of glucose concentration.

*3) Machine Learning Models:* The acquired voltage data was utilized to develop predictive regression models for glucose concentration estimation. Three models were implemented and compared:

- **Linear Regression (LR):** applied as a baseline linear model.
- **Multiple Linear Regression (MLR):** incorporating all three voltage readings as independent variables.
- **Random Forest Regressor (RFR):** an ensemble learning model capable of capturing nonlinear relationships and interactions within the dataset.

*4) Feature Representation:* The feature vector for each sample was defined as:

$$\mathbf{X} = [V_{\text{baseline}}, V_{\text{pre}}, V_{\text{post}}]$$

The target variable was the corresponding glucose concentration (mg/dL). Model performance was evaluated using standard regression metrics, including root mean square error (RMSE) and Clarke Error Grid analysis to assess clinical relevance.

*5) Mode 2: Photodiode-Based Model Architectures:* The photodiode-based dataset used voltage features as input, consisting of: baseline voltage ($V_{\text{baseline}}$), pre-beam voltage ($V_{\text{pre}}$), and post-beam voltage ($V_{\text{post}}$). Three regression models were evaluated, summarized in Table IV.

## IV. IMPLEMENTATION DETAILS

### A. SWIR Imaging Operating Mode

*1) Region-Based CNN Model Training Protocol:* We selected the best-performing architecture, the Region-Based CNN (Model M4), implemented using Python with TensorFlow and Scikit-Learn. Models were trained independently for each SWIR wavelength. All images were resized to $128 \times 128$ pixels and their intensities normalized to the $[0, 1]$ range. We used a 70:30 train-test split with a fixed random seed for reproducible evaluation on the unseen test set.

*2) Experimental Workflow:* Each model was trained for 50 epochs with a batch size of 32. We used the Adam optimizer with Mean Squared Error (MSE) as the loss function and monitored Mean Absolute Error (MAE).

### B. Voltage Reading Mode

*1) Random Forest Regression Details:* For the photodiode voltage recordings, Random Forest Regression (RFR) provided the most accurate glucose concentration estimates, as it effectively handles the non-linear relationships in the voltage signals.

*2) Model Configuration:* The RFR model, implemented in `scikit-learn`, was configured with 100 estimators and a maximum depth of 15 to prevent overfitting. Pre-beam voltage refers to the baseline voltage across the photodiode before the light source is activated, while post-beam is the measured voltage during illumination. Feature scaling was performed using min-max normalization to ensure consistent input ranges across the ensemble learners.

## V. RESULTS

### A. Overview

Table V presents a full comparative summary of model performances across different wavelengths and preprocessing strategies. The models are identified according to the architecture codes introduced earlier. All reported metrics were computed on the unseen test sets using the evaluation framework described in Section III.

The voltage-based photodiode approach achieves its lowest error when employing nonlinear models. The Random Forest

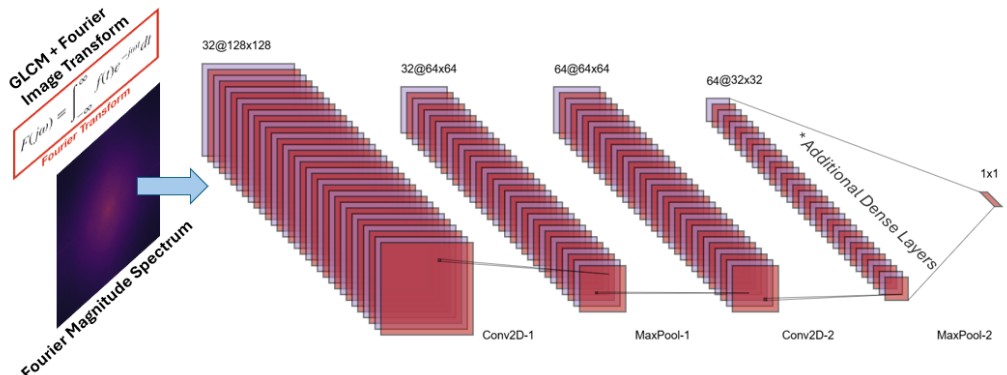

Fig. 5: Architecture of the feature extraction pipeline. The raw image undergoes Fourier transform followed by GLCM computation to extract texture features (contrast, homogeneity, energy, and correlation) for regression modeling.

TABLE III: Summary of Model Architectures

| Ref. | Model | Architecture Summary |
|------|-------|----------------------|
| M1 | 3-layer MLP | Fully connected MLP with 3 hidden layers, `tanh` activations, and dropout (0.3) after each layer. |
| M2 | 4-layer MLP (Swish) | Fully connected MLP with 4 hidden layers, `Swish (SiLU)` activations, batch normalization. |
| M3 | Hybrid Funnel Network | Shrinking-layer architecture with Swish activations, increasing $L_2$ regularization depth-wise, and dropout. |
| M4 | Region-Based CNN | Shallow CNN with max-pooling and dropout, followed by dense layers for regression; see Fig. 5. |

TABLE IV: Summary of Photodiode-Based Model Architectures

| Ref. | Model | Architecture Summary |
|------|-------|----------------------|
| LR | Linear Regression | Ordinary least squares using a single voltage feature to predict glucose concentration. |
| MLR | Multiple Linear Regression | Linear model using $V_{baseline}$, $V_{pre}$, and $V_{post}$ as predictors. |
| RFR | Random Forest Regressor | Tree ensemble using all voltage features with bagging and random feature selection per split. |

Regressor (RFR) achieved a MAPE of 14.1% after post-/pre-beam normalization, clearly outperforming simpler linear methods such as Linear Regression (LR) and Multiple Linear Regression (MLR). The linear models failed to capture the optical signal's nonlinear dependencies, exhibiting correlation coefficients below 0.1. Therefore, further analysis is required to verify reproducibility.

In contrast, SWIR imaging with convolutional feature extraction via a Region-Based CNN achieved a MAPE of 4.82% at 650 nm, demonstrating superior spatial information use. However, CNNs demand substantial computational resources and hardware, potentially limiting deployment in low-power wearables.

Photodiode systems, while less accurate, remain attractive for ambulatory monitoring due to their small size, low cost, and minimal power requirements when coupled with robust nonlinear regressors.

### B. Wavelength Optimality Analysis

To determine the most effective wavelength for CNN-based glucose prediction, we evaluated performance metrics across four illumination conditions: 650 nm Laser, 808 nm Laser, 850 nm LED, and 850 nm Laser. Fig. 6 summarizes the Root Mean Squared Error (RMSE), Mean Absolute Error (MAE), and Mean Absolute Percentage Error (MAPE) for each wavelength.

As shown in Fig. 6, the lowest RMSE, MAE, and MAPE values at 650 nm indicate that this wavelength provides the

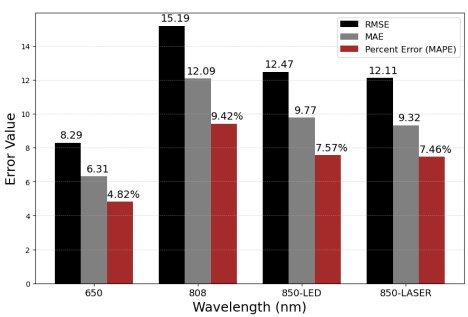

Fig. 6: Error metrics (RMSE, MAE, MAPE) for CNN-based glucose prediction at different wavelengths.

highest signal-to-noise ratio for imaging glucose-dependent spectral changes in our setup. Both 850 nm conditions (LED and Laser) achieve intermediate error rates; however, the 850 nm Laser slightly outperforms the 850 nm LED in terms of MAE and MAPE, suggesting that coherent illumination at 850 nm yields marginally better feature extraction over 808 nm for our CNN.

Overall, these results demonstrate that 650 nm laser illumination is optimal for CNN-based in vitro glucose prediction under our experimental conditions, yielding the lowest prediction error variability with a standard deviation of 9 mg/dL and an average inference time of 426 ms. This low standard deviation indicates that the model's predictions are not only

TABLE V: Summary of Model Performance Across Techniques, Preprocessing Strategies, and Wavelengths

| Mode | Pre-Processing | Model | Wavelength | RMSE | MAE | MAPE |
|---|---|---|---|---|---|---|
| **SWIR Imaging** | Gray-Level Co-Occurrence Matrix & Fourier Domain | 3-Layer MLP (**M1**) | 650 | 15.32 | 12.20 | 10.5% |
| | | | 808 | 19.71 | 15.70 | 13.64% |
| | | | 850$^{avg}$ | 17.26 | 13.80 | 11.8% |
| | | 4-Layer MLP (**M2**) | 650 | 13.83 | 10.30 | 9.55% |
| | | | 808 | 16.70 | 11.70 | 10.56% |
| | | | 850$^{avg}$ | 18.62 | 14.85 | 13.05% |
| | | Hybrid Funnel (**M3**) | 650 | 8.36 | 6.30 | 5.47% |
| | | | 808 | 19.46 | 15.59 | 13.21% |
| | | | 850$^{avg}$ | 11.27 | 8.22 | 7.31% |
| | Convolution-Based Features Extraction | Region-Based CNN (**M4**) | 650 | 8.29 | 6.31 | **4.82**% |
| | | | 808 | 15.19 | 12.10 | 9.42% |
| | | | 850$^{avg}$ | 12.33 | 9.32 | 7.52% |
| **Voltage Readings** | Post-Beam to Pre-Beam Ratio | Linear Regression (**LR**) | 1600 | - | - | - |
| | | Multiple Linear Regression (**MLR**) | 1600 | - | - | - |
| | | Random Forest Regressor (**RFR**) | 1600 | 17.62 | 14.05 | 14.07% |

accurate on average but also consistent across samples. Consequently, we adopt 650 nm as the primary wavelength for subsequent model refinement and integration into the final glucometer design.

*C. Clarke Error Grid Analysis*

The Clarke Error Grid (CEG) was employed to provide a clinically relevant evaluation of the glucose prediction accuracy across all experimental conditions. We present the results for each approach individually (Figs. 7a–8), followed by a comparison with prior studies.

*1) CNN-Based Image Mode:* CNNs were trained independently on image data under different wavelength and illumination settings. Figs. 7a–7d show the CEG plots:

- **650 nm Laser**: 100.0% in Zone A.
- **850 nm LED**: 98.5% Zone A, 1.5% Zone B.
- **850 nm Laser**: 94.2% Zone A, 4.2% B, 1.6% D.
- **808 nm Laser**: 91.5% Zone A, 8.5% Zone B.

Linear baselines from Javid et al. [22] failed to generalize under our 2 mg/dL resolution. We attribute this to their coarser sampling, which masked the limits of linear models under finer-grained conditions.

*2) Photodiode-Based Voltage Mode:* Fig. 8 shows the CEG for the RFR model on 1600 nm photodiode voltage ratios:

- **RFR (1600 nm LED)**: 86.4% Zone A, 13.6% Zone B.

While trailing CNN performance, this mode remains clinically viable and offers a compact, cost-effective alternative for real-time glucose monitoring.

*3) Comparison with Existing Work:* Our best Zone A results (100.0% at 650 nm, 98.5% at 850 nm) exceed those of Cistola et al. [11], despite their simpler aqueous glucose setup. By incorporating synthetic skin and chromatic blood mimics, our system better approximates real-world conditions and confirms the robustness of both CNN and RFR models under clinically relevant constraints.

## VI. Conclusion and Future Work

A CNN-based imaging system achieved a mean absolute percentage error (MAPE) of 4.82% at 650 nm with 100% Zone A coverage on the Clarke Error Grid, while a compact photodiode-voltage sensor reached 86.4% Zone A accuracy. These two techniques were benchmarked independently, demonstrating distinct strengths in terms of clinical accuracy, cost-effectiveness, size, and power efficiency. By leveraging spatial and spectral information, our study provides meaningful progress beyond traditional invasive methods and previously reported non-invasive optical techniques.

This work provides a proof-of-concept validation before human clinical trials. Unlike prior studies with simple aqueous glucose solutions, we created a synthetic blood phantom that better replicates physiological optical conditions. These results establish a strong basis for future in vivo validation, where human-specific variables—such as skin thickness, hydration, temperature, skin tone, and biomarkers like PPG, HRV, and BMI—will be examined

Future research will validate these models on diverse human subjects under real-world conditions, considering motion artifacts, temperature changes, and perspiration. We also plan to implement adaptive calibration to improve robustness against physiological variability. From an engineering standpoint, we will optimize the CNN pipeline for edge deployment, reduce inference latency, and explore hybrid strategies that fuse imaging and voltage data at runtime. Collectively, these efforts aim to enable a reliable, wearable, continuous glucose monitoring system for everyday clinical use.

## VII. Acknowledgment

This work was supported in part by the NIH/NIA (Award P30AG073105) and by the Georgia Research Alliance (Award GRA.25.016.KSU.01.a). The content is solely the authors' responsibility and does not necessarily represent official NIH views.

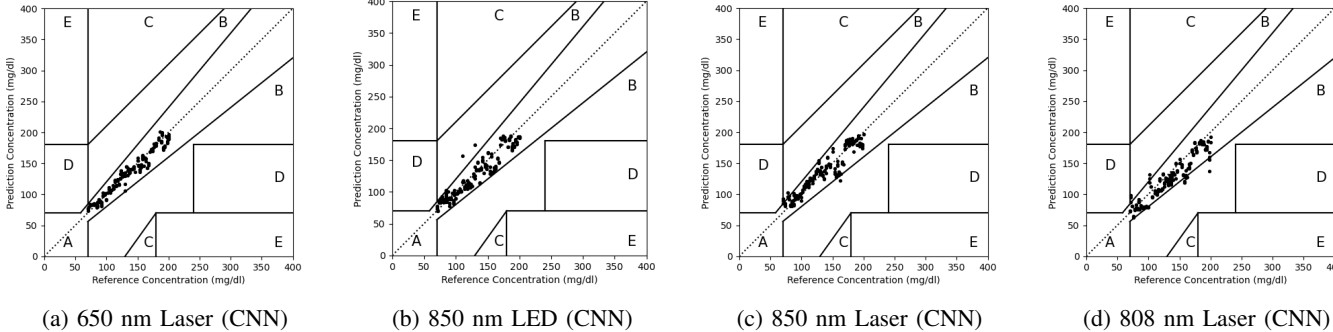

(a) 650 nm Laser (CNN)  (b) 850 nm LED (CNN)  (c) 850 nm Laser (CNN)  (d) 808 nm Laser (CNN)

Fig. 7: Clarke Error Grid results for the CNN model at (a) 650 nm Laser, (b) 850 nm LED, (c) 850 nm Laser, and (d) 808 nm Laser.

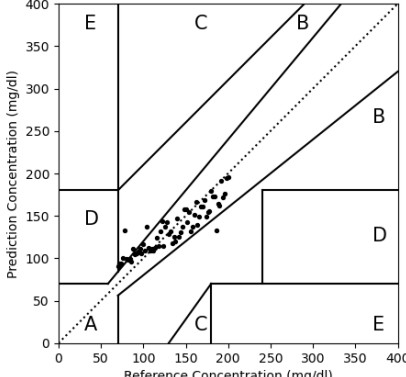

Fig. 8: Clarke Error Grid for photodiode-based voltage mode (1600 nm LED, RFR) – 86.4% Zone A.

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
