# OpenReview forum: "Reliable Noninvasive Glucose Sensing via CNN-Based Spectroscopy"
_IEEE.org/EMBS/BHI/2025/Conference — BHI 2025_

### Official Review · Reviewer_wZUd · 2025-06-29
**Review of "Reliable Noninvasive Glucose Sensing via CNN-Based Spectroscopy"**

**Confidence:** 4
**Clarity Of Writing:** good
**Clinical Significance:** great
**Methodological Novelty:** good
**Overall Rating:** 5
**Final Rating:** 6

**Experiments And Results:**

fair

**Questions For The Authors:**

1. What ablation studies were conducted to justify the selection on the feature extraction with GLCM and DFT?
2. Can the authors provide insights into the source of errors or failure cases?

**Strengths:**

1. Dual-modal approach is novel in combining imaging and voltage sensing for providing practical deployment options.
2. Quantitative results of the CNN-based model, with 4.82% MAPE and 100% Zone A Clarke Error Grid coverage, are strong compared to prior work.
3. Methodology is well-structured and clearly presented with clear motivation.

**Summary Of The Paper:**

In this paper, the authors introduce a dual-modal AI framework for non-invasive glucose monitoring using short-wave infrared (SWIR) spectroscopy with both CNN-based imaging and photodiode voltage sensing approaches. The main contribution is achieving 4.82% MAPE using CNN model on multi-wavelength SWIR images, based on feature extraction with Gray Level Co-Occurrence Matrix (GLCM) and discrete Fourier Transform (DFT) to enhance the model's ability to capture complex patterns.

**Weaknesses:**

1. The paper is limited on validation scope. Without any validation on real human subjects, or any equivalent evaluation of key physiological variables (skin tone, thickness, hydration, temperature), the experimental data raises concerns about its applicability to real world settings. The relatively small SWIR Imaging Dataset (6,500 samples) further limits generalizability claims.
2. Discussion of computational requirements or inference times of the proposed methods are missing. This is important for assessing the real-world applicability of the two methods.

---

### Official Review · Reviewer_iCnx · 2025-07-13
**This paper presents two AI-based frameworks for non-invasive glucose sensing using SWIR spectroscopy, using CNN-based image analysis and photodiode voltage regression. While the imaging modality achieves high accuracy on synthetic phantoms, the work lacks real-world validation, overstates deployability, and omits key baseline results. The contribution is technically promising but not yet ready for acceptance without needed revisions.**

**Confidence:** 4
**Clarity Of Writing:** good
**Clinical Significance:** great
**Methodological Novelty:** good
**Overall Rating:** 4
**Final Rating:** 7

**Experiments And Results:**

poor

**Questions For The Authors:**

1. In III.B.4 why are texture or frequency patterns expected to change due to glucose concentration and so the two types of features are chosen?
2. Why the results of LR and MLR were not good enough compared to random forest?
3. As the real-world data is not ideal and noisy, what would be the performance of the proposed solutions in real-world settings?

**Strengths:**

The paper addresses an important medical challenge with real-world impact.
Diabetes affects hundreds of millions of people, and current glucose monitoring methods (like finger-pricking) are painful, costly, or inconvenient. The authors focus on a painless, non-invasive alternative, which could notably improve quality of life and patient compliance.

It proposes two complementary sensing methods, offering flexibility: The study explores both a high-accuracy imaging system and a low-power photodiode system. These two proposed solutions can cover different types of systems and needs.

Unlike simpler studies that use just water or glucose solutions, the authors use synthetic blood and skin-like silicone layers. This creates more realistic light scattering and absorption conditions, making the experiment conditions closer to real-world scenarios.

The paper is easy to follow and has been written clearly.

**Summary Of The Paper:**

his paper presents a dual-mode AI system for measuring blood sugar levels non-invasively (without pricking the skin), using light-based sensing and machine learning.

Two separate methods are developed and compared:

1. SWIR Imaging Mode – Uses a special camera to capture near-infrared images of fake blood samples. These images are analyzed using deep learning (CNN) to predict sugar levels.

2. Photodiode Voltage Mode – Uses a small light sensor that measures how much light passes through the same samples. The readings are analyzed using a simpler machine learning model (Random Forest).

The system was tested using synthetic skin and blood samples with glucose concentrations ranging from 70–200 mg/dL. The CNN-based image system gave very accurate results (4.82% error, 100% clinical safety), while the photodiode system gave moderate accuracy (14.07% error, 86.4% clinical safety) but with lower hardware requirements.

Together, these two systems offer a trade-off: one prioritizes accuracy, the other portability, aiming for future wearable glucose monitors that are low-cost and practical for real-time use.

**Weaknesses:**

Despite calling the framework “dual-modal,” the two sensing methods are completely independent. They were trained and evaluated separately. There is no model fusion, joint learning, or combined decision-making, which is usually expected from a dual-modal system. The authors should rename the framework to avoid this problem.

My main concern regarding the results is that all experiments are done on artificial samples. The models haven’t been tested on human subjects or real skin, since factors like sweat, noises, and temperature can significantly alter performance. This makes the generalization unproven.

In III.B.4 the authors have introduced two types of features, gray level and Fourier features, but there should be added a justification on why each feature type was added supported by prior works.

I like the reasons justifying the choice of the 1600 nm photodiode wavelength such as compact design for mobile health and compact design for mobile health and Exploration of a less-studied region (1600 nm) in literature, but the authors should investigate alternative wavelengths, combine multiple wavelengths, or engineer more complex features beyond raw voltage readings. As a result, the photodiode system’s full potential would remain under-explored. Its lower accuracy might be improved with further exploration. The authors should preferably add a section exploring other features or add this as a limitation of this work.

The imaging system is presented as suitable for wearable integration. But as the CNN model requires full image processing, Fourier transforms, and GLCM computation, to prove they are suitable, there should be a real-time computation and hardware requirements section containing inference time, CPU and GPU usage, and power draw.
Wearables require lightweight models. Without runtime data, it's unclear if this system can ever work on embedded hardware.

In Table IV, the authors omit results for both LR and MLR in the photodiode mode. While they briefly state these models underperform compared to Random Forest in V, they provide no justification for not having predicted performance (an analysis of the distribution of the data and the predictivity features of these models and analyzing why they fail to predict values in this case, and also the exact numbers, so readers can observe the gaps.

---

### Official Review · Reviewer_7NHN · 2025-07-17
**Promising dual‑modal SWIR framework with strong in‑vitro accuracy, but limited real‑world evidence and incomplete statistical/benchmarking**

**Confidence:** 4
**Clarity Of Writing:** great
**Clinical Significance:** good
**Methodological Novelty:** good
**Overall Rating:** 6

**Experiments And Results:**

fair

**Questions For The Authors:**

- Do you plan to test on human subjects soon? This could improve clinical relevance.

- Can you report confidence intervals or SDs for MAPE and Zone A to strengthen the result credibility?

- What are the CNN’s compute and memory needs? Can lightweight models maintain accuracy?

- Your baseline comparisons (e.g., with Cistola et al. [11]) use different datasets, limiting direct performance comparison. Have you considered re-implementing such models, including recent deep learning–based approaches, on your dataset to enable fair benchmarking and better validate your performance claims?

**Strengths:**

- High CNN accuracy (MAPE 4.82%, 100% Zone A) at 650 nm with feature fusion improves robustness (Table V, Fig. 7a).

- Dual-modal system offers a balanced trade-off between accuracy and deployment cost (Section III, Fig. 1).

**Summary Of The Paper:**

The paper proposes a dual-modal non-invasive glucose monitoring system using SWIR imaging and photodiode voltage sensing. The imaging mode employs CNNs with GLCM and DFT feature fusion, achieving a 4.82% MAPE and 100% Zone A coverage at 650 nm. The photodiode-based voltage mode, using machine learning regressors, achieves 14.1% MAPE and 86.4% Zone A. All evaluations are on synthetic phantoms.

**Weaknesses:**

- No in-vivo validation limits real-world applicability (Section VI).

- Performance reported without SD or confidence intervals (Table V).

- Photodiode mode underperforms significantly compared to CNN (14.1% MAPE, 86.4% Zone A vs. 4.82%, 100%) (Table V, Fig. 8).

---

### Official Review · Reviewer_cT4k · 2025-07-17
**Analysis of blood glucose estimation based on SWIR / transmission intensity**

**Confidence:** 4
**Clarity Of Writing:** good
**Clinical Significance:** fair
**Methodological Novelty:** fair
**Overall Rating:** 4

**Experiments And Results:**

good

**Questions For The Authors:**

- Have the authors investigated the attention mechanism as an alternative to naive concatenation?
- How do the authors explain the model performing best with illumination in the visible range?
- No validation on anything other than the phantom. Is the phantom adequate? What is the composition of synthetic blood and synthetic skin?
- Does the SWIR  band information improve the features meaningfully even when illumination is  visible range?

**Strengths:**

- Multispectral SWIR spectroscopic problem formulation as a means for non-invasive blood glucose estimation
- Promising metrics
- Reasonably well written and organized

**Summary Of The Paper:**

The paper proposes Convolutional Neural Network based regression of blood glucose levels given texture/frequency domain feature maps extracted from Short-Wave Infrared images. Additionally, the paper includes analyses of photodiode-based estimation of blood glucose using a Random Forest Regressor to serve as a simpler input modality.

**Weaknesses:**

- The paper would benefit from a clearer description of what pre-beam and post-beam measurements are, including the method of sampling each - beam splitter, dichroic mirror, etc. assuming they are voltages corresponding to the energy of the beam before and after interaction with the phantom.
- Proposed method does not include calibration. This is mentioned in future work, however.
- Phantom does not replicate clinical conditions - Cuvette path length would be about an order of magnitude longer that a vessel diameter. The experimental setup would have a significantly larger diameter and consequently large-footprint cross-sections and would also be homogenous unlike in vivo conditions with vessels being much thinner than the musculoskeletal structure they are embedded in.
- Including the composition of the synthetic blood and skin would be pertinent.
- 650nm (visible range) illumination with a SWIR camera yielding the best results is not motivated/justified adequately. How would it compare with a visible range camera?
- Documenting the input and target dimensions would be expected for any trainable method.
- Light source power description missing.
- An image of the actual system rather than a 3D model would be ideal.

Not quite a weakness of the proposition itself but the preprint for this paper is available on ArXiV which I stumbled upon in the course of inspecting references. Author information is not hidden there and consequently I am aware of author names, compromising the double-blind nature of this review in principle, despite my not knowing the authors or having a direct competing interest.